# The Wooden Roof Framing Elements, Furniture and Furnishing of the Etruscan *Domus* of the *Dolia* of Vetulonia (Southern Tuscany, Italy)

Ginevra Coradeschi [1], Massimo Beltrame [1], Simona Rafanelli [2], Costanza Quaratesi [2], Laura Sadori [3] and Cristina Barrocas Dias [1,4,*]

1    HERCULES Laboratory, University of Évora, Largo do Marquês de Marialva, n.8, 7000-804 Évora, Portugal; ginevrac@uevora.pt (G.C.); massimo@uevora.pt (M.B.)
2    Isidoro Falchi Civic Archaeological Museum, Piazza Vetluna, 58043 Vetulonia, Italy; simrafan@gmail.com (S.R.); costanza.quaratesi@gmail.com (C.Q.)
3    Department of Environmental Biology, Sapienza University of Rome, Piazza Aldo Moro, n.5, 00185 Rome, Italy; laura.sadori@uniroma1.it
4    Chemistry Department, School of Sciences and Technology, University of Évora, Rua Romão Ramalho, n.59, 7000-671 Évora, Portugal
*    Correspondence: cmbd@uevora.pt

**Abstract:** The Etruscan *Domus* of the *Dolia* remained hidden until 2009, when archaeological excavations began in the Etruscan–Roman district of Vetulonia (Southern Tuscany). Based on the classification of the archaeological materials recovered, the destruction of the *Domus* and the Etruscan city of Vetulonia was traced back to the 1st century BC. The highly various and precious materials recovered inside the *Domus* revealed the richness of the building and its inhabitants. With this study, we present the anthracological analyses from the *Domus* of the *Dolia*. Wood charcoals were recovered from different house rooms, which had different functions based on the archaeological evidence. The tree species employed for the construction of the roof of the building were deciduous and semi-deciduous oak wood (*Quercus* sect. *robur*, *Quercus* sect. *cerris*) and silver fir wood (*Abies* cf. *alba*). Evergreen oak wood (*Quercus* sect. *suber*), boxwood (*Buxus sempervirens*), beech wood (*Fagus* cf. *sylvatica*), maple wood (*Acer* sp.) and cherry wood (*Prunus* cf. *avium*) were adopted for the furniture and furnishings of the house. Moreover, wood charcoal fragments of fruit trees belonging to the family of Rosaceae were identified, documenting a possible garden inside the court of the house. The study shows the use of the local tree species primarily. The silver fir wood and beech wood were likely sourced from the nearby (roughly 60 km) Mount Amiata.

**Keywords:** archaeobotany; etruscan archaeology; Southern Tuscany; charcoal; wood exploitation

## 1. Introduction

The Etruscans, known as *Tyrsenoi* or *Thyrrenoi* by the ancient Greeks, flourished in central Italy between the 9th and the 1st century BC. They mainly settled and ruled in central Italy (in part of the present-day territory of the Tuscany, Lazio and Umbria regions), with important presences in Campania, in a large part of the Po valley (Emilia Romagna, Lombardy, Veneto) and on Corsica Island. Etruscans were one of the most important civilizations of ancient Italy, and the funerary equipment recovered on several monumental burial mounds is widely known, being evidence of the degree of their civilization. Organized through a confederation of 12 city states (*Dodecapoli*), they are quite famous for their deep metallurgical skills, for the ability of their artisans in gold processing, for their capability of trading, and for their cult of the deaths. Moreover, they were also skilled sailors and traders, having contact with the most important civilizations of the Mediterranean Area. From the 3rd century BC, the Etruscan civilization was slowly adsorbed by the Roman Empire (three of the legendary kings at the dawn of ancient Rome were Etruscans) [1–4].

Even if much is known about their cult of death, little is known, on the contrary, on Etruscan daily life and settlements [5]. In particular, the exploitation of natural resources for construction purposes has rarely been addressed [6,7], and few data are available regarding the exploitation and selection of wood [8–13].

Different scholars ascribed to Etruscans, through pollen analysis, the responsibility of forest clearance that heavily altered the surrounding environment [14–17], but systematic archaeobotanical studies from Etruscan settlements are occasional, and a clear connection between the opening of forests and the wood used for building, cooking and smelting has never been established.

Given the rarity of this type of study, the analysis of the charred woods of the *Domus* of the *Dolia* proves to be important, in particular for the deepening of knowledge about the Etruscan world of the living, which still today remains largely unknown.

The discovery of the *Domus* of the *Dolia*, located in the ancient town of Vetulonia (Castiglione della Pescaia, Southern Tuscany, Italy), and the study of the materials retrieved, made it possible to create an accurate reconstruction of the entire Etruscan house, and to describe the life and the activities of the wealthy inhabitants of the *Domus* between the 3rd and the 1st century BC [18,19]. Moreover, the extraordinary nature of the architectural data unearthed by the archaeological excavation allowed for the acquisition of information concerning the city of Vetulonia in the last centuries of its Etruscan history. Among the materials retrieved from the excavation of the *Domus,* there were bricks, mortars, roof tiles, different ceramic wares, votive bronze statues, coins, nails, and numerous charred woods, with these being the last the object of this study. The analysis of wood charcoals belonging to several structural and furnishing elements coming from different rooms of the *Domus* of the *Dolia* represents the first study regarding the choice and the use of wood by the Etruscans of Vetulonia.

Thus, the objectives of this study are:

1.  To hypothesize the probable origin of the plants used through the identification of the timber at the family/genus/species level.
2.  To understand the reasoning behind the wood species selection for construction purposes through the evaluation of the wood-related technological knowledge of the Etruscan carpenters.

Overall, this study aims to shed light on the technological, economic and social aspects of the inhabitants of the *Domus* of the *Dolia,* and, in a sense, of the Etruscan community of Vetulonia.

## 2. Archaeological Settings

### 2.1. The Etruscan City of Vetulonia and the Discovery of the Domus of the Dolia

Vetulonia is located in Southern Tuscany, and it was an important city during Etruscan times. It ruled in a vast territory, extremely rich in natural resources such as metals, mined in the so-called area of the Colline Metallifere. During the first centuries of its Etruscan history (the 9th–11th centuries BC) it was a vital centre, full of artisan shops, well known for its bronze workers and goldsmiths [20]. Between the 8th and the 6th centuries BC, Vetulonia was the most important of the 12 Etruscan city states, and it survived to the Roman expansion of the 2nd century BC. Nevertheless, it was forced to become an ally of Rome, maintaining his own identity. This was a period of prosperity for the city. The beginning of the Etruscan Vetulonia decline was the result of its involvement in the Roman civil war, which led to the destruction of the city in the 1st century BC [21].

The name of the ancient town of Vetulonia had disappeared from official documents in 1201. The first archaeological works, which lasted from 1882 to the early stage of the 1900s, were conducted by Dr. Isidor Falchi [22]. Falchi discovered the funerary area of the city, discovering both the *tombe a pozzetto* (pit tombs) from the Iron Age—the Villanovan phase of Etruscan Culture (end of the 9th and the beginning of the 8th centuries BC)—and the majestic *tumulus* tombs, attributable to the so-called Etruscan Orientalizing Period (from the 8th century to 580 BC) [23,24]. Moreover, the excavations led also to the discovery of

the ancient Etruscan city which, with the royal decree of 22 July 1887, regained the ancient Etruscan name of Vetulonia [22]. The excavations conducted by Falchi were concentrated in the area known as Poggiarello Renzetti, where the Etruscan–Roman district of the city was brought to light (Figure 1).

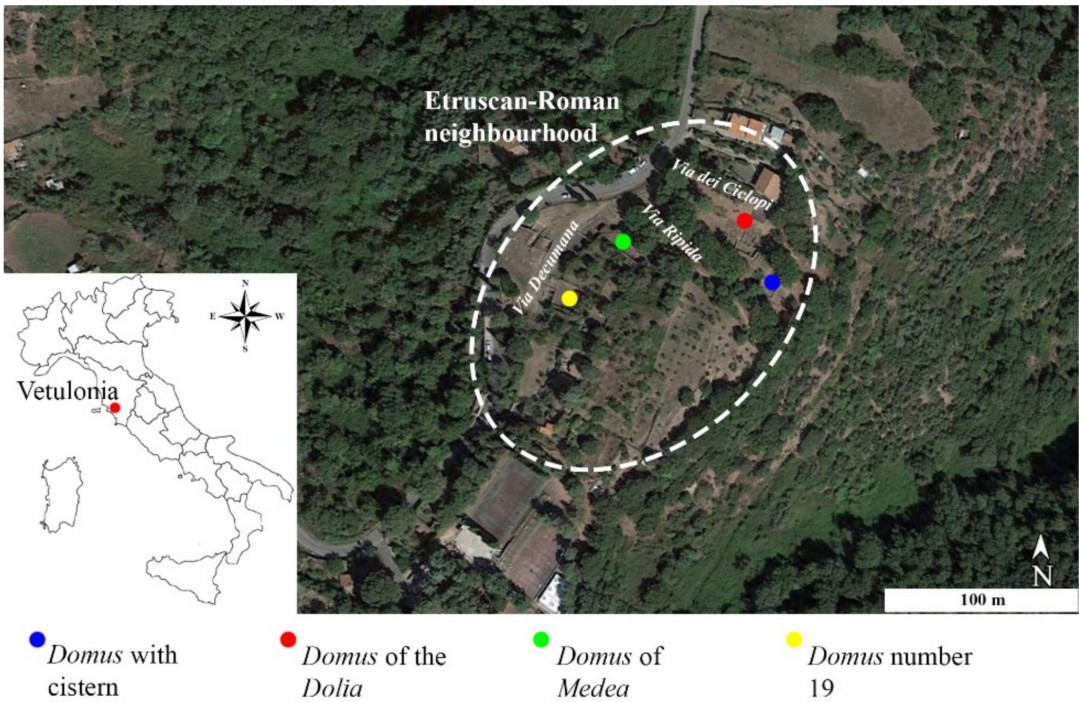

**Figure 1.** The Etruscan district of Poggiarello Renzetti. Source: Google Earth Pro.

The area was the subject of new archaeological excavations in late 1980 by the superintendent Mario Cygielman, who discovered the *Domus* of *Medea* [25]. Other archaeological evidence of the urban Etruscan area was excavated by Anna Talocchini between 1960 and 1970, referring to the sanctuary part of the city (i.e., the *Acropolis*). These are the areas of Costia Lippi and Costa Murata, both chronologically attributable to the Hellenistic (4th–3rd centuries BC) and to the Late Republican periods (2nd–1st centuries BC), respectively [26,27]. In addition, some evidence (i.e., pottery) suggested that the area of Costa Murata was already occupied (in the second half of the 6th and the middle of the 5th centuries BC) during the archaic period [28]. Furthermore, the well-known Mura dell'Arce, first discovered by Isidoro Falchi at the end of 1800, which—according to new research—refers to a chronological period between the 3rd and the 2nd centuries BC, are worth mentioning [23,28]. Finally, in 2009, under the direction of Simona Rafanelli (current director of the Etruscan Museum of Vetulonia), a new archaeological excavation took place in the Poggiarello Renzetti area (the Etruscan–Roman neighborhood discovered by Falchi), and the *Domus* of the *Dolia* was brought to light [29].

The archaeological area of Poggiarello Renzetti represents the richest area of Etruscan Vetulonia settlements. The district is crossed by a large, paved street, called via Decumana, which overlooks numerous shops and private houses. In the north-eastern part of the main road network there are two perpendicular streets, respectively known as the via Ripida and the via dei Ciclopi [30,31]. It is within the insula defined by these three paths that the *Domus* of the *Dolia* was discovered. The *Domus* supports other residential Etruscan structures of Hellenistic period (4th–1st centuries BC) such as the *Domus* n. 19, the *Domus* of *Medea*, and the *Domus* characterized by the presence of a vast atrium with *impluvium* and of an adjacent large cistern carved into the rock [31–33], (Figure 1). The Hellenistic quarter of Poggiarello Renzetti represents, therefore, with its important housing structures, the

most striking archaeological evidence of the revival of Vetulonia to new splendour with a building and economic renaissance, which started from the 3rd century BC.

*2.2. Archaeological Evidence of the Domus of the Dolia*

The *Domus* of the *Dolia* was destroyed by a fire. The classification of the recovered archaeological materials, such as black-painted ceramic and Greek–Italic amphorae, suggest a chronology between the 3rd and the 1st centuries BC. It is known that the Etruscan city was destroyed by the troops of Lucio Cornelio Silla in the 1st century BC, in the aftermath of the victory over Gaius Mario's army during the Roman civil war [34].

This *Domus* represents an exceptional discovery for the archaeological area of Vetulonia and for Etruscan archaeology, as well-preserved dwellings with high rises (over 1.60 m and about 6 cm in thickness) have rarely been found. Moreover, because of the fire, the roof collapsed, sealing and preserving an old context with a wide variety of materials.

The building is divided into about 12 rooms (the excavation is still underway) (Figure 2), and from the excavation data (ongoing study) it seems to have had three construction phases. Phase I (beginning of the 3rd century BC) seems to have included five different rooms (A, C, E, G and D) arranged to the south. Rooms E and G seem to have been interconnected, and Room D probably served as a house entrance at this stage. From the II phase on (2nd century BC), Room E was separated from Room G, becoming one of the most important rooms of the house. From this phase, the *Domus* seemed to be arranged around a big semi-open courtyard—Room D—which, after being refurbished, become the first atrium of the house, possibly with an *impluvium*. Furthermore, during this phase, Rooms F and H (a courtyard with a portico) and room B (product processing area) were likely built. During phase III (between the 2nd and the beginning of the 1st century BC) the *Domus* was expanded to the north with the addition of other rooms (e.g., P, S) and a large peristyle (K). During this phase of the house's life, Room E seems to have replaced the function of Room C, becoming the most important room of the house. The *Domus* was configured like that until its destruction [34].

Many precious remains were recovered inside the *Domus,* revealing the richness of this building and its inhabitants. From one of the most significant rooms of the house, the formal dining room, the *Triclinium* (Room C), archaeologists recovered a small white limestone column, a precious bowl decorated with an animal head, and a few bronze coins, possibly associated to a foundation ritual related to the new life-phase of the house [29]. This room was also characterized by painted walls with red and blue frescos belonging to the first Pompeian style [35]. The *Domus* also included a representative guest reception room, the *Tablinum* (Room E), with a beautiful floor decorated with a meander motif, formed by white and grey limestone tiles and plaster (*Opus Signinum* style). The walls of this room were also decorated with frescoes [29,35]. Inside the two storage rooms (Rooms A and G) many amphoras and big earthen pots (*Dolia*) were discovered, one of which is about 1.20 m high and is still almost intact. The uniqueness of these discoveries justifies the name of the *Domus*. Moreover, in the storage room G a small treasure was also discovered, consisting of some votive bronzes. The house also included a room perhaps associated with grape or olive processing (Room B) [34]. The architecture of the *Domus* seems to refer to a specific type of rural aristocratic housing with a peristyle court, diffuse in the Attic region from the 5th century BC, called *Pastas* style. This type of dwelling is well documented in the southern part of the Italian peninsula during the Hellenistic age [36,37].

The material employed for the construction of the *Domus* of the *Dolia* was sandstone, which was extracted for the construction of the perimeter walls. In some rooms, the walls were probably finished by using raw clay material. Tiles and brick tiles were used to cover the roof, while raw bricks were employed in the structure of some of the internal divisions of the house. Wood was utilized for the construction of the room's framing elements, for the doors and maybe for the internal support of the raw clay parts of the walls, which were probably built with a different kind of technique [29]. It is important to report that

no wooden remains associated with the building process of the raw clay walls have been retrieved. Moreover, the few raw clay remains bring back the negative of small stem reeds.

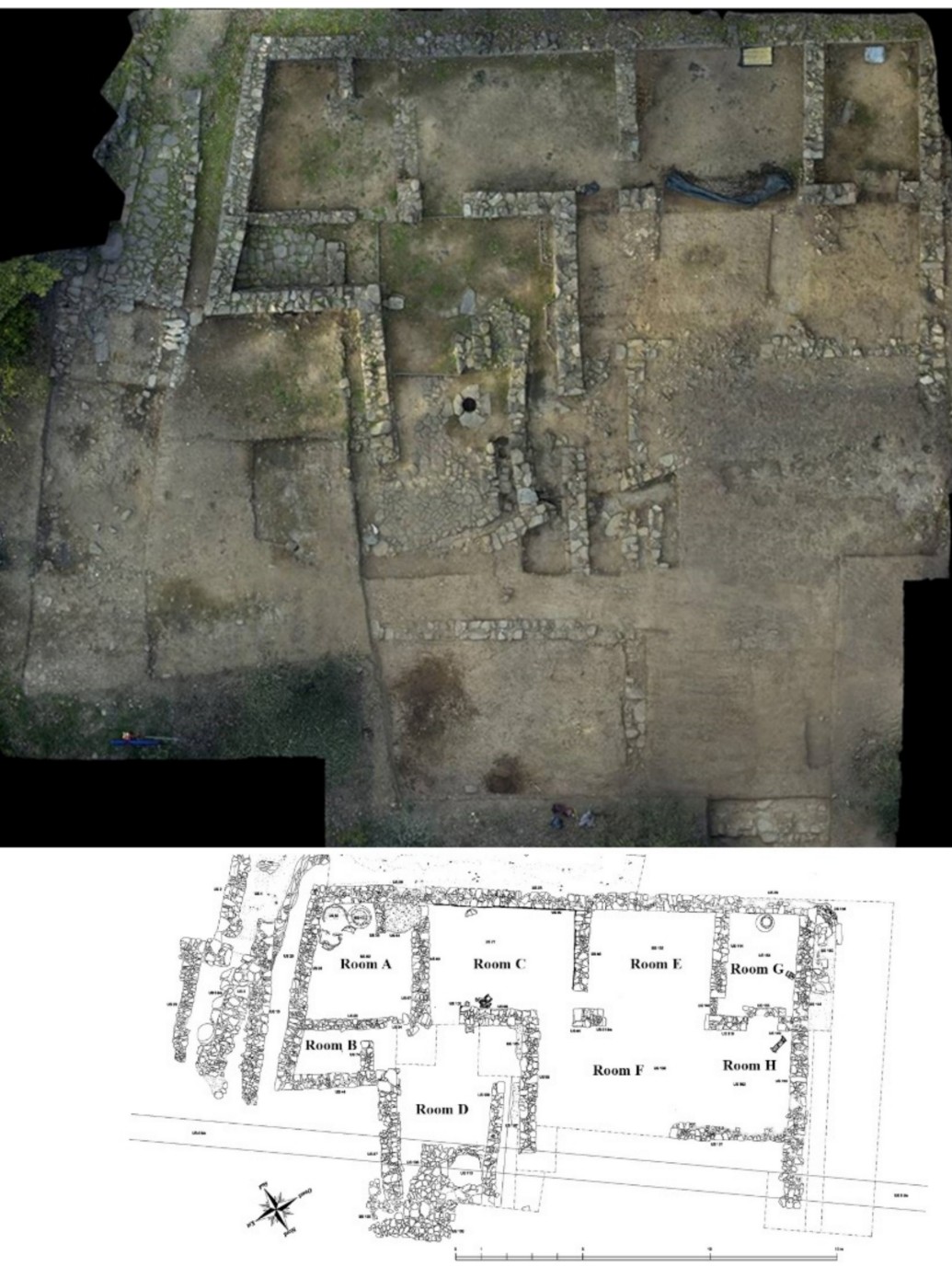

**Figure 2.** *Domus* of the *Dolia* divisions, pictures by Paolo Nannini and Stefano Spiganti.

From the archaeological data and from the *Domus* typology, we can assume that the house was supplied with light from the central courtyard; therefore, the presence of windows in the investigated area of the house is not expected. The huge *Domus* of the *Dolia*, containing several compartments with various well-preserved and precious materials uncovered within the wood charcoal remains (i.e., the object of this study) represents a unique case for Etruscan archaeology.

## 3. Materials and Methods

### 3.1. Materials and Sampling

The materials analysed comprise wood charcoal remains belonging the first archaeological campaigns (2011–2016) of the *Domus* of the *Dolia*. These archaeological campaigns were conducted within the better-preserved part of the house, the southern part, which was the most relevant investigated area due to the presence of collected wood charcoal evidence. The northern part of the house (excavation ongoing) was largely destroyed by agricultural works, and the archaeobotanical evidence is scarce or absent. The wood charcoal remains under study are representative of the II and III rebuilding phases of the *Domus*, which was ultimately ruined and buried under the fire. The charred wood samples come from seven different compartments of the house, namely Room A (storage), Room C (*Triclinium*), Room D (semi-open courtyard–first atrium), Room E (*Tablinum*), Room F (court), Room G (storage) and Room H (semi-open room with portico) (Table 1; Figure 2).

Anthracological analyses were not included in the initial planning of the archaeological excavation. For this reason, most of the wooden material under study was identified visually, collected by hand (in small and restricted areas), drawn and documented. This material consisted of large charred wooden elements, generally broken/fragmented (the largest fragments were 5 cm in diameter), that were interpreted by the archaeologists as wooden roof framing elements based on their size, position and context during the excavation (Table 1; Figure 3).

**Table 1.** *Domus* of the *Dolia*. The provenance and number of the charcoals analysed, and their archaeological ascription (SU stratigraphic unit).

| Room | SU | Sample Excavation Data | | | No. of Wood Charcoal Fragments |
|---|---|---|---|---|---|
| | | Classification/Information | Charred Wooden Elements | Wood Charcoal from Dark/Black Areas | |
| **A** | 61 | Wooden roof framing elements | 2 | | 20 |
| | 62 | Wood charcoals | | x | 60 |
| **C** | 77 | Wood charcoals-close to nails | | x | 9 |
| | 98 | Wood charcoals | | x | 48 |
| | 102 | Wood charcoals-close to nails | | x | 80 |
| **D** | 111 | Wooden roof framing element | 1 | | 10 |
| **E** | 128/129 | Wood charcoals | | x | 71 |
| | 129 | Wood charcoals-close to nails | | x | 55 |
| **F** | 166 | Wood charcoals | | x | 34 |
| **G** | 143 | Wooden roof framing elements | 3 | | 30 |
| | 143 | Wood charcoals | | x | 50 |
| | 143 | Wooden roof framing element | 1 | | 10 |
| | 143 | Wood charcoals with plaster–near the collapse | | x | 65 |
| | 143 | Wood charcoals–corner of the room | | x | 27 |
| | 145 | Wood charcoals from the interior of a *Dolium* | | x | 26 |
| | 145 | Wood charcoals | | x | 25 |
| | 146 | Wooden roof framing element–corner of the room | 1 | | 10 |
| | 146 | Wooden roof framing elements–close to a *Dolium* | 2 | | 20 |
| | 147 | Wood charcoals-close to nails | | x | 50 |
| | 151 | Wooden roof framing element | 1 | | 10 |
| | 151 | Wood charcoals–close to amphorae and nails | | x | 315 |
| | 152 | Wooden roof framing element | 1 | | 10 |
| | 152 | Wood charcoals–centre of the room/near votive bronzes | | x | 150 |
| **H** | 159 | Large and elongated wooden piece | 1 | | 10 |
| | | **Total** | | | **1195** |

However, it must be considered that most of the wooden components of the *Domus* were destroyed by the fire and by the collapse of the roof; the attribution of the material, during the excavation, to a single original wooden element was not always possible. These uninterpreted wood charcoal remains were generally dispersed in wider dark/black areas of the archaeological layers. Attempts were made to collect the entirety of the remains in these darkened areas, but some smaller charcoal fragments may have been lost. Within the category of wood charcoal from the dark/black areas, one should also consider the possibility that remains from furniture and different furnishings/objects of the *Domus* could also be present in the samples recovered for analysis.

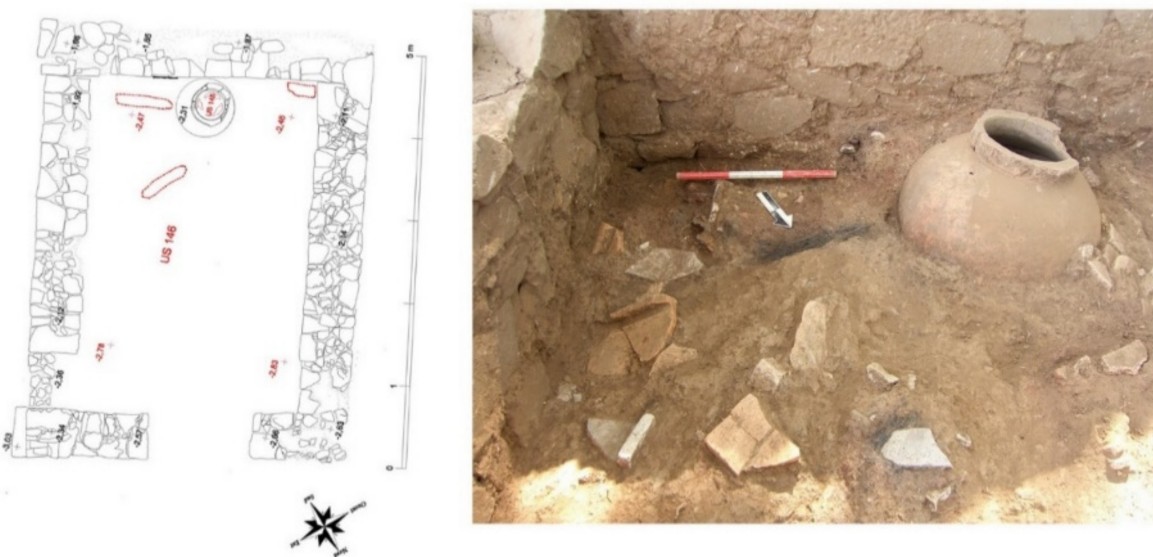

**Figure 3.** Drawing and a picture of the different wooden roof framing elements identified in Room G. Picture and drawing by Stefano Spiganti.

From the material retrieved from the excavation, and given its fragmentary nature, the sampling of the wood charcoal remains was organised in the following way: for each charred wooden element identified during the archaeological excavation, 10 of the largest fragments were selected in order to confirm their provenance from the same tree species; the samples not identified during the excavation phase (samples from the dark/black areas) were studied in their totality.

### 3.2. Anthracological Analysis

The charcoal fragments were manually fractured to expose the three diagnostic sections (transversal, tangential and radial) under a stereo-zoom microscope (LEICA M205C equipped with a camera). The charcoal's anatomical features were determined on a Reflect Light Optical Microscope (LEICA DM2500 equipped with a camara) at different magnifications (5–100x). A selection of charcoal fragments was analysed with a Scanning Electron Microscope (SEM) to obtain high-resolution images. Analyses were performed using a variable pressure HITACHI S3700N SEM, and the operating conditions for the analysis were as follows: secondary electron mode (SE), 10 kV accelerating voltage, 10 mm working distance, 65 µA emission current and <1 Pa pressure in the chamber. All of the samples were covered with a gold layer prior to the analysis. Wood atlases [38–41] were used as comparative tools for the charcoal identification, together with an in-lab reference collection of wood specimens. The analysis identifies wood charcoals fragments at the highest possible taxonomical level (family/genus/section/species). The level of identification depends, in fact, on the available and visible diagnostic micro-anatomical characters of single wood species/specimens. Some other types of observation were also made. For samples identified as building elements, in fact, when possible, the growth ring

curvature was observed [42–45]. When present, galleries of xylophagous insects were also recorded [43,46].

For most of the charcoal samples, the identification of the species was possible, although in some cases the degradation of the wood only allowed for the identification of the genus or the family. In this case, the name of the genus is followed by the abbreviation of species (sp.), i.e., *Acer.* sp. When the species attribution is highly probable, the abbreviation (cf.) is placed between the name of the genus and the name of the species, i.e., *Prunus* cf. *avium.*

Regarding the identification of the wood belonging to the genera *Abies* (fir wood) and *Fagus* (beech wood), their anatomy does not allow any distinction between the species [41,47], but considering the investigating period and the present-day distribution, they can be ascribed to *Abies alba* and *Fagus sylvatica*, respectively.

For the distinction in *Quercus* (deciduous, semideciduous and evergreen sections) we followed the indication of Cambini [38]. These guidelines distinguish the deciduous oaks (*Quercus* sect. *robur*) from the semideciduous (*Quercus* sect. *cerris*) and the evergreen (*Quercus* sect. *suber*). In Italy, the deciduous oak group includes *Quercus robur* L., *Quercus pubescens* Wild., *Quercus frainetto* Ten. and *Quercus petrea* (Matt) Liebl.; the semideciduous include *Quercus cerris* L., Quercus *trojana* Webb and *Quercus aegilops* L.; the evergreens include *Quercus suber* L., *Quercus ilex* L. and *Quercus coccifera* L.

*3.3. Quantification*

Regarding the quantification analysis, it is generally advisable to previously determine which may be the most useful and relevant method to be used in a particular occurrence [48–50]. In this case study, two different quantitative methods were employed: the frequency and the ubiquity correction [51,52]. The frequency (%), based on the absolute number of charcoal fragments, was employed in order to evaluate the different categories of wood charcoals of which the provenience was attributed over the context of the *Domus*–wood roof framing elements, furnishing and court tree/s. The ubiquity correction (%U) was used to show the occurrence of a tree species across the *Domus* of the *Dolia* contexts.

These methods considered only wood charcoal fragments coming from the dark/black areas. An assessment based only on the quantification of the *taxa* identified as single wood elements during the excavation would potentially obscure the importance of some *taxa* as building material.

**4. Results**

In total, 1195 wood charcoal fragments coming from seven different compartments of the house—rooms A, C, D, E, F, G, H—were analysed and identified.

A total of nine *taxa* were identified. Amongst them were *Abies* cf. *alba, Acer* sp., *Buxus sempervirens, Fagus* cf. *sylvatica, Prunus* cf. *avium, Quercus* sect. *cerris, Quercus* sect. *robur, Quercus* sect. *suber* and Rosaceae (Table 2, Figures 4 and 5).

The most exploited woods for the construction of the roof of the *Domus* of the *Dolia* were deciduous and semideciduous oak and silver fir wood (Table 2). Deciduous and semideciduous oak wood accounts for the highest number of wood charcoal fragments identified; silver fir wood is the most recurrent taxon across the different rooms of the *Domus* (Table 3). Wood roof framing elements account for 66.57% of the total of the charcoal fragments analysed. Evergreen oak wood, boxwood, beech wood, maple wood and cherry wood were used for the furniture of the house and the furnishing objects of the house. Furnishing woods account for 30.23% of the total. Charcoal fragments belonging the Rosaceae family were probably part of a tree/s of the garden of the court of the house, accounting for 3.19% of the total (Figure 6).

**Table 2.** Plant source and possible interpretation (SU stratigraphic unit).

| Room | SU | Interpretation Based on the Archaeological and Archaeobotanical Data | Plant Source | No. of Wood Charcoal Fragments |
|------|-----|------|------|------|
| **A** | 61 | Roof beams or rafters | *Quercus* sect. *robur* | 20 |
| | 62 | Fragments of wooden roof beam/s or rafters/s | *Quercus* sect. *robur* | 15 |
| | 62 | Fragments of wooden roof beam/s or rafter/s | *Abies* cf. *alba* | 45 |
| **C** | 77 | Fragments of wooden roof beam/s or rafter/s | *Quercus* sect. *cerris* | 9 |
| | 98 | Fragments of wooden furniture | *Fagus* cf. *sylvatica* | 48 |
| | 102 | Fragments of wooden roof beam/s or rafter/s | *Quercus* sect. *cerris* | 30 |
| | 102 | Fragments of wooden roof beam/s or rafter/s | *Abies* cf. *alba* | 50 |
| **D** | 111 | Roof beam or rafter | *Quercus* sect. *cerris* | 10 |
| **E** | 128/129 | Fragments of a *Klíne* bed foot | *Buxus sempervirens* | 71 |
| | 129 | Fragments of wooden roof beam/s or rafter/s | *Abies* cf. *alba* | 55 |
| **F** | 166 | Fragments of fruit tree/s of the courtyard | Rosaceae | 34 |
| **G** | 143 | Roof beams or rafters | *Quercus* sect. *robur* | 30 |
| | 143 | Fragments of wooden roof beam/s or rafter/s | *Quercus* sect. *robur* | 50 |
| | 143 | Roof beam or rafter | *Abies* cf. *alba* | 10 |
| | 143 | Fragments of wooden roof beam/s or rafter/s | *Abies* cf. *alba* | 65 |
| | 143 | Fragments of wooden tool | *Acer* sp. | 27 |
| | 145 | Fragments of wooden roof beam/s | *Quercus* sect. *robur* | 25 |
| | 145 | Fragments of a jar lid of a *Dolium* | *Prunus* cf. *avium* | 26 |
| | 146 | Roof beams or rafters | *Quercus* sect. *robur* | 30 |
| | 147 | Fragments of wooden roof beam/s or rafter/s | *Quercus* sect. *robur* | 50 |
| | 151 | Roof beam or rafter | *Quercus* sect. *robur* | 10 |
| | 151 | Fragments of wooden roof beam/s or rafter/s | *Quercus* sect. *robur* | 315 |
| | 152 | Roof beam or rafters | *Quercus* sect. *robur* | 10 |
| | 152 | Components of a container/support of votive bronzes | *Quercus* sect. *suber* | 150 |
| **H** | 159 | Dividing door of rooms G and H | *Prunus* cf. *avium* | 10 |
| | | **Total** | | **1195** |

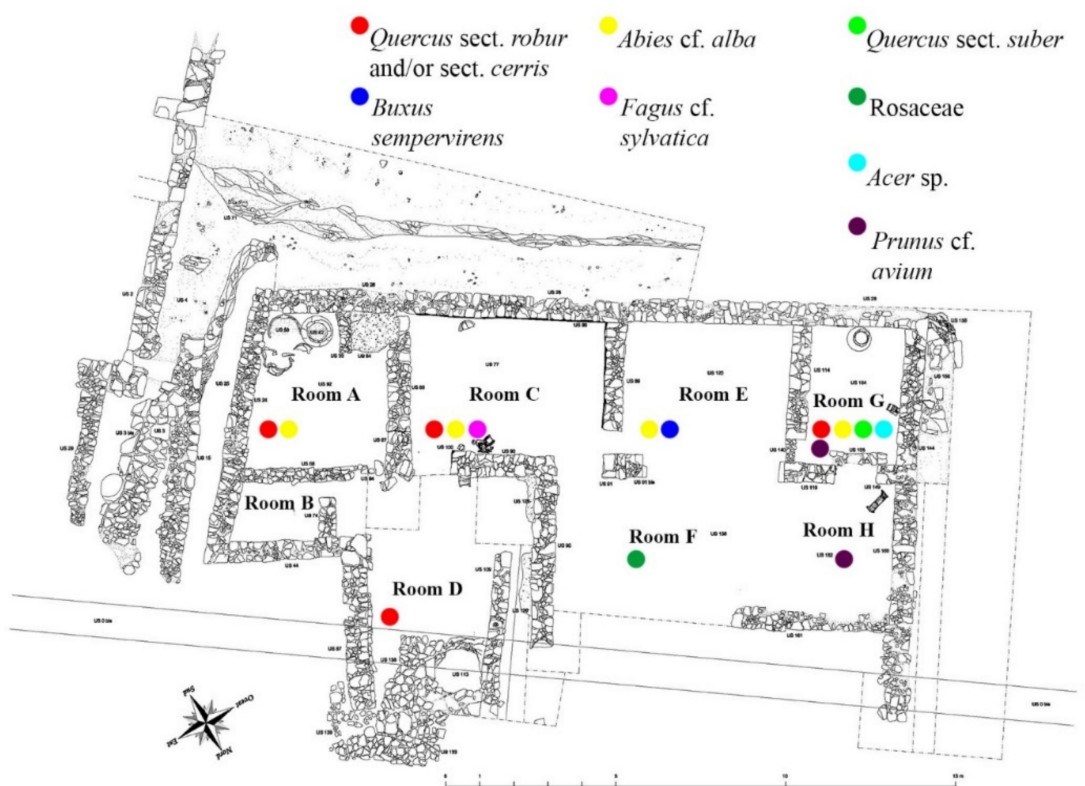

**Figure 4.** Plant source distribution in the different rooms of the *Domus* of the *Dolia*.

**Table 3.** Plant source occurrences in the different divisions of the *Domus* of the *Dolia* (samples from the dark/black areas).

| | | | | Plant Source | | | | | |
|---|---|---|---|---|---|---|---|---|---|
| | *Quercus* **sect.** *robur* and **sect.** *cerris* (Deciduous and Semideciduous Type) | *Abies* cf. *alba* | *Quercus* **sect.** *suber* (Evergreen Type) | *Buxus sempervirens* | *Fagus* cf. *sylvatica* | Rosaceae | *Acer* sp. | *Prunus* cf. *avium* | |
| Room | No. of Wood Charcoal Fragments | | | | | | | | Total |
| A | 15 | 45 | | | | | | | 60 |
| C | 39 | 50 | | | 48 | | | | 137 |
| E | | 55 | | 71 | | | | | 126 |
| F | | | | | | 34 | | | 34 |
| G | 440 | 65 | 150 | | | | 27 | 26 | 708 |
| Total | 494 | 215 | 150 | 71 | 48 | 34 | 27 | 26 | 1065 |
| % | 46.38 | 20.19 | 14.08 | 6.67 | 4.51 | 3.19 | 2.54 | 2.44 | 100.0 |
| %$U_t$ | 23.12 | 32.87 | 4.24 | 11.27 | 7.01 | 20 | 0.76 | 0.73 | 100.0 |

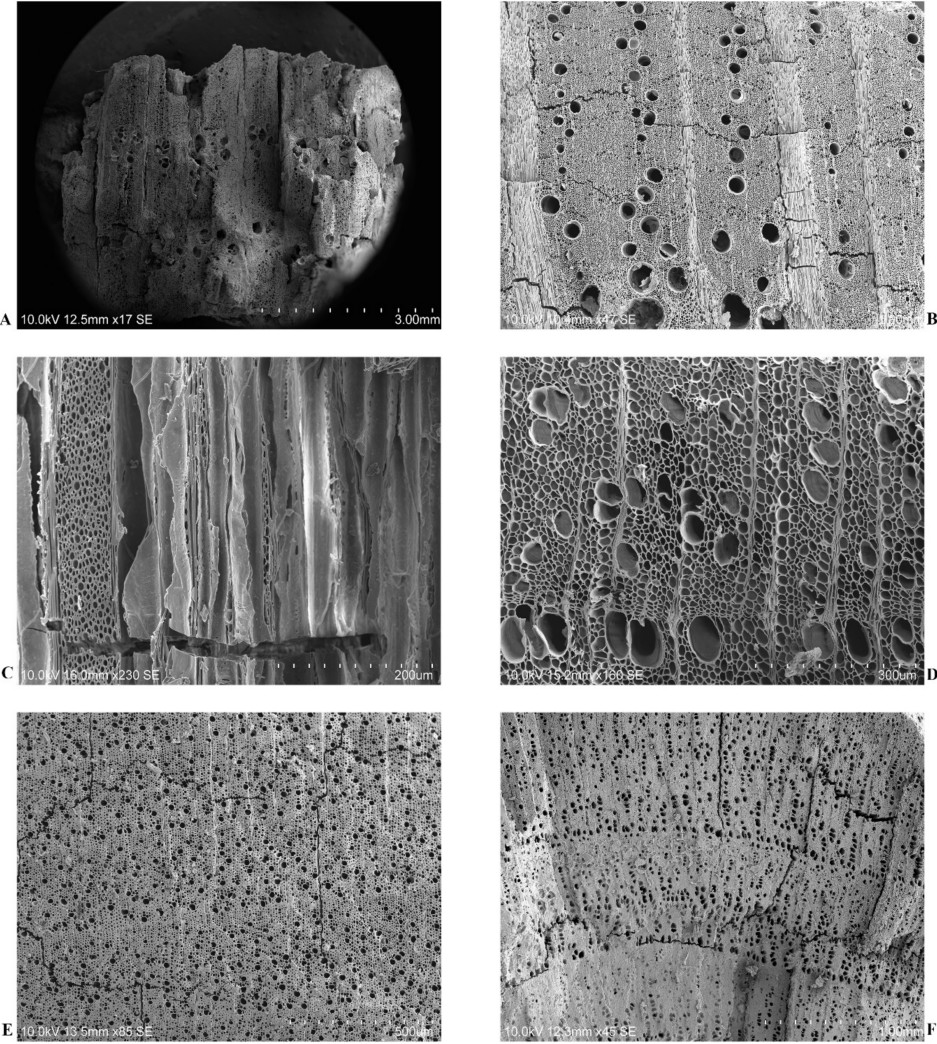

**Figure 5.** Transversal section of *Quercus* sect. *robur* (**A**); transversal section of *Quercus* sect. *cerris* (**B**); tangential section of *Fagus* cf. *sylvatica* (**C**); transversal section of *Prunus* cf. *avium* (**D**); transversal section of *Buxus sempervirens* (**E**); transversal section of Rosaceae (**F**).

Regarding the tree ring observation, fragments interpreted as wooden roof framing elements (including those identified during the excavation and those from the dark/black areas) it was not possible to identify a curvilinear trend of the tree growth rings. Tyloses were observed in the lumen of the spring wood vessels of 350 samples identified as *Q.* sect. *robur*. Galleries formed by xylophagous insects were observed in 85 samples of *Q.* sect. *robur*, seven samples of *Q.* sect. *cerris*, 174 samples of *A.* cf. *alba* and 23 samples of *F.* cf. *sylvatica* (Figure 7). The presence of bark was not detected for any of the samples under study.

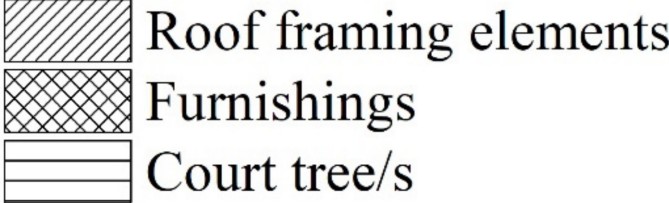

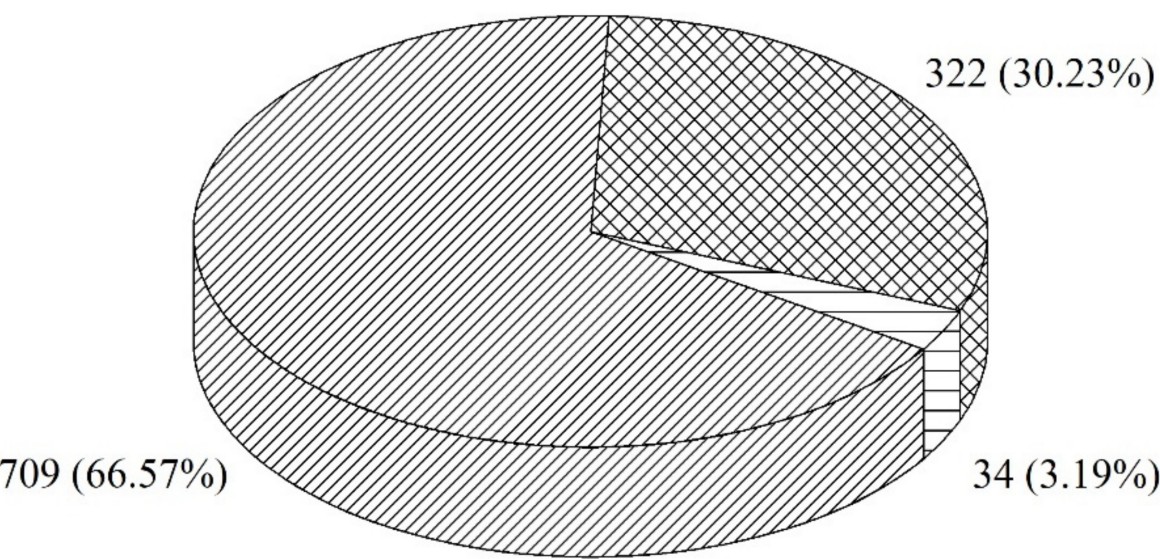

**Figure 6.** Percentages of the roof framing elements, furnishings and court tree/s identified.

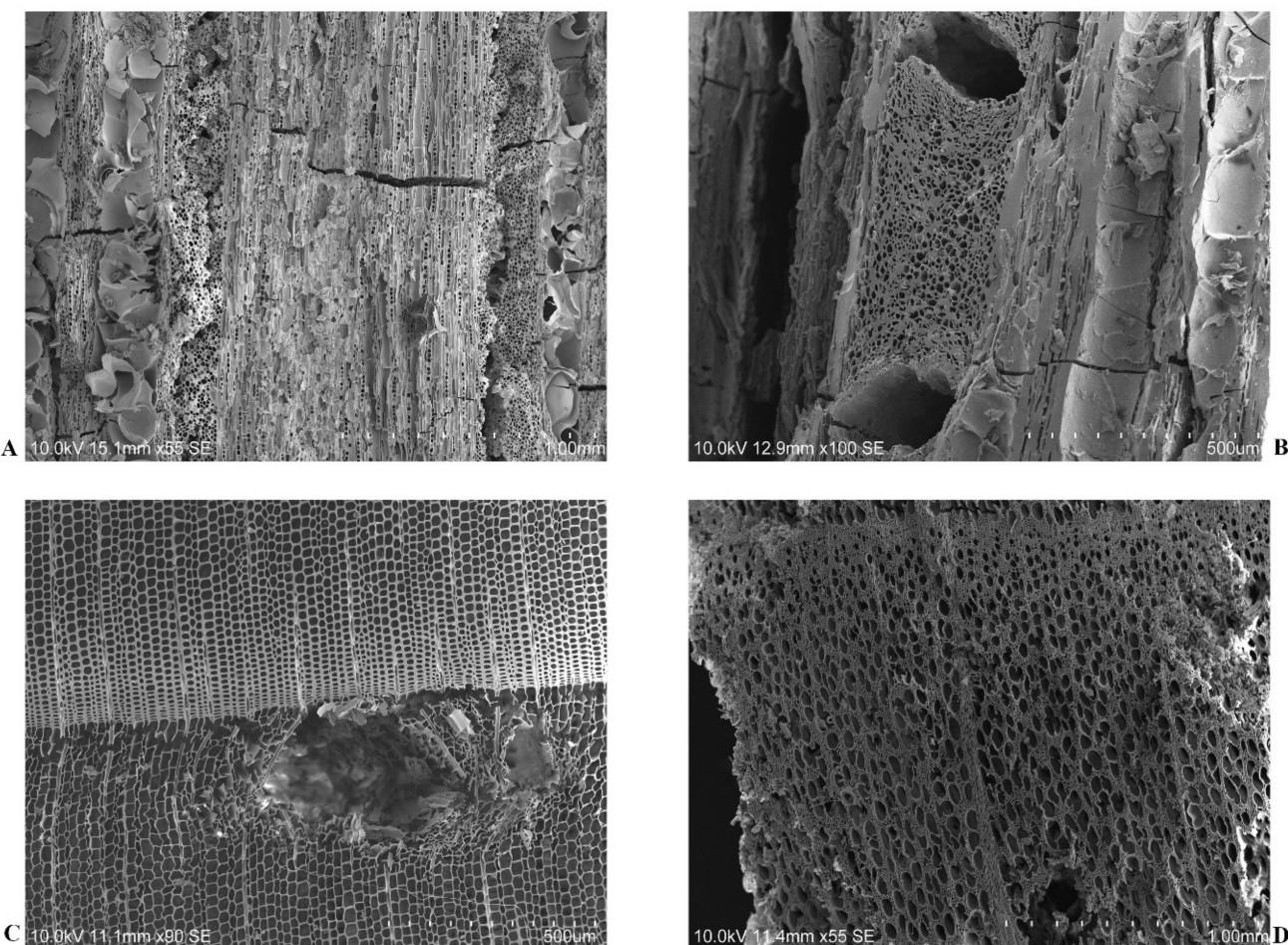

**Figure 7.** Tyloses in the lumen of the spring vessels of a *Quercus* sect. *robur* fragment (tangential section) (**A**); xylophagous'
galleries affecting a *Quercus* sect. *cerris* fragment (tangential section), *Abies* cf. *alba* fragment (Transversal section) and *Fagus*
cf. *sylvatica* fragment (Transversal section) (**B**–**D**).

## 5. Discussion

### 5.1. Different Wood Species for Different Uses

#### 5.1.1. Wooden Roof Framing Elements

Relating to the charred woods identified, those employed for the construction of the
*Domus* of the *Dolia* were *Q.* sect. *robur* (deciduous type), *Q.* sect. *cerris* (semideciduous
type) and *A.* cf. *alba* (silver fir wood) (Table 2). The wood charcoal samples belonging
to these species were interpreted as the beams or rafters which made up the roof of
the *Domus*. As previously reported, some of these wooden elements were identified
during the archaeological excavation. In these cases, the taxa identification supported the
archaeological interpretation. The archaeological data, along with the state of conservation
of these wooden remains, did not allow any distinction between the beams and/or the
smallest rafters. In any case, the traces and the position in the archaeological layers of some
of these elements made it possible to understand the type of roof covering some parts of
the house. Concerning the four rooms aligned in the southern part of the house (A, C, E,
G), the covering was a single pitch roof. The roof pitch was angled from above to below,
where the compluviate rooms were located. In the case of the wood charcoals found in
the dark/black areas, the preservation state of the samples and /or their burial conditions
did not allow any interpretation during the excavation phase. In these circumstances,
the interpretation was possible due to a careful study of the excavation data and of the
physical and mechanical characteristics of the identified woods. The comparison with other
archaeological realities (e.g., where the same species of woods had also been employed for

the same purposes) was also useful for their interpretation. Numerous charcoal samples were retrieved in association with many nails bent at right angle (Table 1). The discovery of the nails used to fasten the beams/rafters was a further clue which allowed for the interpretation of these wooden elements.

The deciduous and semideciduous oak wood (i.e., *Q.* sect. *robur* and *Q.* sect. *cerris*) was exploited for the manufacturing of the roof beams and/or of the rafters of rooms A, C, D and G. Eleven roof beams/rafters made of this wood were identified during the archaeological excavation as single elements, whilst other wood remains of this species were retrieved in a fragmentary state (a sampling of the charred woods coming from the dark/black areas) (Table 1).

This type of oak wood is well appreciated for its properties, especially for the mechanical strength and the durability of its heartwood, as its high content of tannins preserves it from biological attacks [53]. Vitruvius and Pliny praised the qualities of this wood [54]. Its use for the manufacture of structural elements has been well attested since prehistoric times, and oak forests cover large geographical areas included in the Mediterranean basin [55]. Numerous studies have attested to the use of oak wood for building during Italian prehistory [56–63], as well as for the historical periods [64,65]. The widespread use of deciduous and semideciduous oak wood for the beams, rafters, columns and boards of many Roman buildings, as well as for the construction of naval frames, demonstrate the extent to which Romans appreciated this wood for construction purposes [66–70]. Concerning the Etruscan world, the most interesting parallel is the Etruscan Farm of Pian d'Alma (Tuscany) [10].

The identification of one roof beam/rafter of turkey wood from room D (the first atrium of the house with a possible impluvium) indicates the likely existence of a roof cover (at least partial) in the last phase of the life of the *Domus*.

It is likely that fir wood (i.e., *A.* cf. *alba*) was also used for the roof beams and/or for the rafters of several rooms of the *Domus* of the *Dolia*, namely rooms A, C, E and G. It was possible to identify only one individual wooden element of this species during the archaeological excavation, while many remains of this species were retrieved only in a fragmentary state (samples of charred woods coming from the dark/black areas) (Table 1). In this study, samples were identified as likely *Abies alba*, given the wide distribution of this tree in the Italian forests, and in the nearby Mount Amiata. The silver fir is the tallest native tree of the Italian peninsula, reaching 45 m in height. The tall, slender stem makes it the ideal wood for the construction of poles, boards and beams. The large stem diameter of this species makes it suitable for the creation of large wooden products. Furthermore, this type of wood can be easily split and sawed, making it very suitable for the production of boards [71]. The qualities of the silver fir wood were widely appreciated among the Roman and Greek carpenters [72,73]. Classical authors have described its qualities and prestige with respect to other tree species, especially for its exploitation as building elements [74]. Theophrastus wrote about its incredible resistance to deformation when tilted, as in the case of roof beams [73]. Vitruvius describes it as the perfect wood for construction due to its resistance, lightness, workability, and the length of its stem [75]. Livy revealed its essential role for the construction of the Roman naval fleet [73].

Numerous fragments of woods and charcoals of fir wood were retrieved in several Italian archaeological sites, largely identified as building elements, confirming the widespread use of this wood, as indicated by ancient sources. Many vertical poles of some Italian prehistoric pile dwelling were made of fir wood [57,58,76–78]. Several roof beams and rafters of Romans basilicas, temples and private buildings were made of fir wood, as charcoals and wood remains testify [40,54,70,73,79–87]. A large amount of wooden remains of this species related to shipbuilding were also retrieved, testifying to the widespread use of this wood for construction frames [69,88–90]. Finally, wood remains of fir wood were retrieved from the Etruscan Sanctuary of Pyrgi [9].

The large amount of charred wood fragments of *A.* cf. *alba* recovered inside of the *Domus* of the *Dolia*, interpreted as roof framing elements, reinforce the hypothesis that the Etruscans were great exploiters of this wood, as were the ancient Romans and Greeks.

The final phase of the *Domus* of the *Dolia* occupation coincides with increasing Roman control over the Etruscan cities. Roman choices in wood selection became more culturally influential at this time, which is apparent in the evidence of the wood exploitation at this site.

Regarding the compartment E (*Tablinum*), the peculiar use of silver fir wood for the manufacture of the roof beams/rafters could be a deliberate choice made by the constructors, perhaps at the behest of the family of the *Domus* of the *Dolia*. The use of a more valuable wood species would have made the environment of this important room more beloved; this type of room was normally reserved for the reception of guests.

Considering the number of fragments, and their frequency and ubiquity, we can observe that the roof framing elements are the most abundant wood charcoals identified at the *Domus* of the *Dolia*. This data should not be surprising, considering that the woods employed for the construction of the roof of the house were used for all of the rooms of the *Domus*, being present in larger quantities even when the *Domus* was still alive. Between the wooden roof framing components, deciduous and semideciduous oak wood were identified in a larger quantity compared with silver fir wood (46.38 vs. 20.19%), while silver fir wood was the most recurrent taxon across the different rooms of the *Domus* (32.87%U) (Table 3). The abundance of deciduous oak wood is intelligible because oak wood is considered, in this context, a local plant source. The higher ubiquity value of silver fir wood could be explained by considering the quality and the beauty of this wood.

### 5.1.2. Furniture and Furnishing Objects

Among the wood charcoals identified, those that were parts of the *Domus* furniture, furnishings, and other objects (i.e., tools) were: *Q.* sect. *suber* (evergreen oak group), *B. sempervirens* (boxwood), *F.* cf. *sylvatica* (beech wood), *Acer* sp. (maple wood) and *P.* cf. *avium* (likely cherry wood). Wood charcoal remains of these species were retrieved only in a fragmentary state (samples of charred woods coming from the dark/black areas). The study of the excavation data, of the physical and mechanical characteristics of this wood, and the comparison with other archaeological realities made the interpretation of these wood charcoal samples possible.

Regarding the evergreen oak wood remains (i.e., *Q.* sect. *suber*), they were recovered from Room G of the *Domus* of the *Dolia*. The evergreen oak wood employed for the construction of the *Domus* may belong to one of the following tree species: the cork oak (*Q. suber*), the holm oak (*Q. ilex*) or the Kermes oak (*Q. coccifera*), however it was not possible to distinguish between the various species of this group at the micro-anatomical level.

Evergreen oak wood has been well known since antiquity [66], especially by Romans, who largely used it for the manufacture of objects and furniture. This type of wood has also been used for the covering of sophisticated furniture, given its qualities [64,65,81]. Pliny, in his *Naturalis Historia*, praises its pleasant nature and reports various uses, in particular its employment for the manufacture of small-sized objects, tools and handles [91]. He described its use for the manufacture of the famous Citrus Venus table, owned by the emperor Tiberius [92], and also wrote about its high resistance to friction and considered it very suitable for the construction of wheels [73]. Cato recommended the adoption of this wood for the manufacture of agriculture tools' handles [93]. Wood remains of evergreen oak have been retrieved in several archaeological sites, both Roman and Etruscan [10,83]. Wooden remains of this species related to shipbuilding have also been retrieved [67,69].

Wood charcoal fragments of evergreen oak were retrieved from the centre of Room G of the *Domus* of the *Dolia*, in connection with votive bronzes (Figure 8). Wood charcoal remains belonging to this species were only retrieved in a fragmentary state. However, the characteristics of this wood and their sampling position seems to suggest their possible use for the bronze's storage, as they could have been parts of a cabinet or a box containing the bronzes.

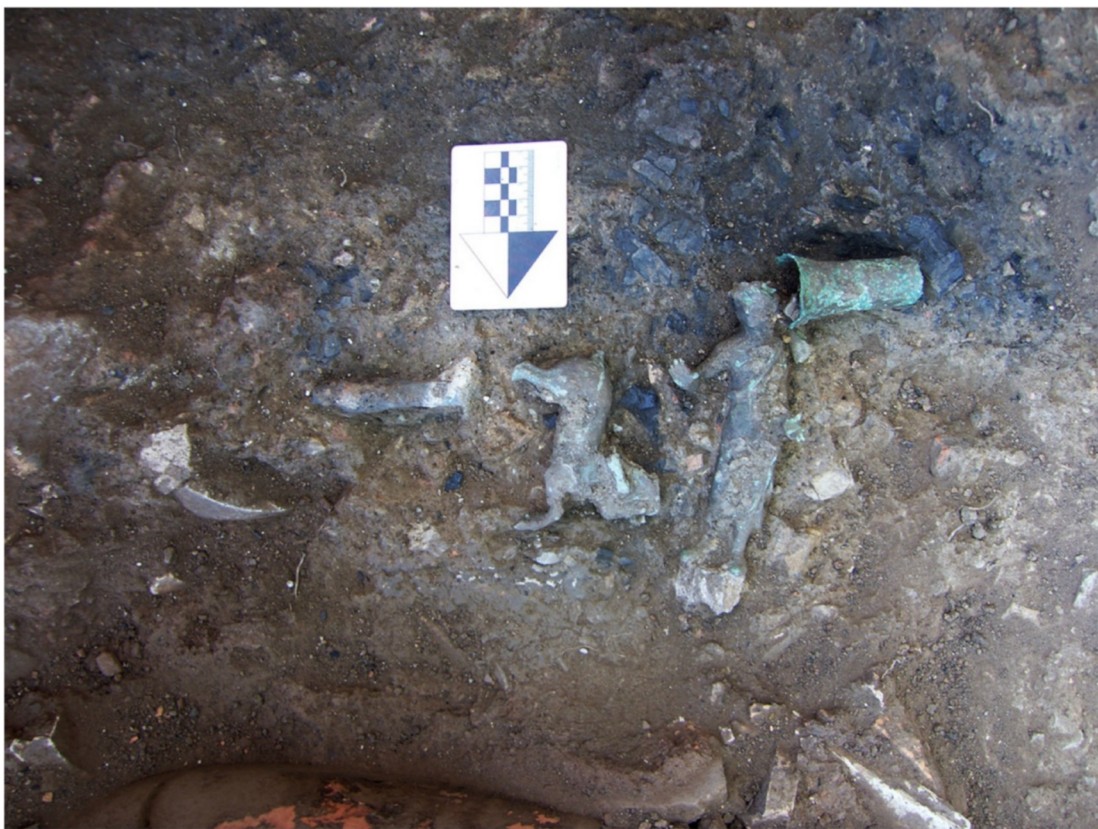

**Figure 8.** Votive bronzes surrounded by charcoals at the time of the discovery.

With regards to the boxwood fragments (i.e., *B. sempervirens*), they were recovered from Room E of the *Domus* of the *Dolia*.

Boxwood is a small shrub between 2 and 4 m high, producing a strong and heavy wood. Its use is associated with the art of cabinet making, carpentry, inlay and turnery works. Its qualities of hardness and resistance to chipping made it the favourite wood for the manufacture of combs since ancient times [73]. This wood was valuable and already appreciated by the Assyrians, who considered boxwood wooden furniture to be the spoils of war [94]. Boxwood was also used in Ancient Egypt and in Italy during the Roman Empire for the manufacture of furniture, tool handles, small precious objects, and wind instruments [95]. This wood has been and still is appreciated for the manufacture of stringed instruments, such as organs and pianos [94]. Moreover, it has always been used for the manufacture of tools, and more generally of objects exposed to wear and friction [55,73]. Boxwood was mentioned in Latin sources, with Pliny describing it as a hard, resistant and hardly perishable wood [91]; Vitruvius described it as suitable for less-visible works, such as clamps supporting false-plastered vaults [73]; Ovid mentioned its use for the manufacture of flutes [73]. Ancient Greek literature mentioned its use for the construction of convivial beds (*Klíne*) [96]. Archaeological remains of boxwood dating back to prehistoric times attested to the wide use of this wood [97–101]. Finally, the boxwood flutes with the tablet recovered from the shipwreck of the Giglio island coast (600 BC) (Tuscany) are from the Etruscan period [8].

Given the characteristics of this wood and the evidence of its use in the ancient literature fragments, the boxwood charcoal samples from room E of the *Domus* of the *Dolia* are likely to be interpreted as part of a foot of a *Klíne* bed, present in this room at the time of the fire in the house. To support this hypothesis, it is important to refer to the fact that the boxwood samples of the *Domus* of the *Dolia* were recovered in association with some bronze elements identified as parts of the foot of a *Klíne* bed. The identification of these bronze elements was confirmed by the comparison with a bronze foot of a Greek *Klíne*

bed currently exhibited at the archaeological museum of Nice. The presence of a *Klíne* bed in the *Tablinum* of the *Domus* of the *Dolia* at the time of the fire is interpreted by the archaeological data as an indication of a second phase of the life of the house, where the *Tablinum* would have replaced the function of the ancient *Triclinium*.

Concerning the wood charcoal fragments of beech wood from Room C of the *Domus* of the *Dolia,* they were identified as likely being *Fagus sylvatica*, given the wide distribution of this tree in the Italian greenwoods and the nearby Mount Amiata.

Beech wood has always been highly appreciated for its strength and versatility [73]. It is an easily worked wood, excellent for being finished and folded; for these reasons, it has always been used in the manufacture of all types of home furniture, but its tendency to crack and bend does not make it a suitable wood for building and construction [40,102]. This wood is highly appreciated for the manufacture of floors, stairs and interiors [103]. It was a well-known wood and widely used by Romans, who employed it for the construction of large, decorated chests that furnished gentlemen's bedrooms [104]. Theophrastus mentioned beech wood as one of the most suitable woods for ship masts, furniture, and especially beds [73]; Pliny described it as an easy wood to work, soft and easily breakable [91]; Columella described chests made of beech wood [92]; Ovid wrote about beech, describing it as the primary source in the manufacture of plates and cups used in the rural houses of the time [73]. Beech was one of the most-used woods at Pompeii and Herculaneum [70,80], where it was also employed for the manufacture of bed legs [92]. Some fragments of beech wood interpreted as parts of a piece of furniture have been recovered from Roman *Domus* and farmhouses [83,105,106]. Components of Roman ships were made with beech wood [67,69].

Wood charcoal samples of beech wood retrieved from Room E of the *Domus* of the *Dolia* could be interpreted as a part of a piece of furniture located in this room. Given the fragmentary nature of this charred wood, it is not possible to make any hypotheses concerning the typology of this furniture. However, given the characteristics of the wood, the context, and the parallels with the ancient literature and other archaeological realities, this hypothesis seems the most likely.

Wood charcoal fragments of maple wood (i.e., *Acer* sp.) were recovered from Room G of the *Domus* of the *Dolia*.

Maple wood has excellent mechanical characteristics; it is easy to work with, it is tenacious and elastic, and easily perishable [40,107]. Its fine texture is useful for turning works [40]. The Greek philosopher Theophrastus mentioned the use of this wood for yokes and bed nets [73]. According to Pliny, it was widely considered a suitable wood for making furniture, and was especially appreciated for tables [73,91,104,108]. The Roman poet Horace also praised the quality of this wood [104]. Charcoals and wood remains of maple wood were retrieved from Italian prehistoric archaeological sites [57,58,109]. For historical periods, we know of the use of this wood by the Egyptians, who particularly appreciated it for the manufacture of chariot frames [64]. Maple wood was widely used by Romans for the manufacture of tools as well as valuable objects, such as mirror frames and musical instruments, as is evidenced by several archaeological remains [83,94,104,110,111]. In the Roman empire, it was also largely used as furniture covering, thanks to the beauty of its wood [64]. During the medieval period it was employed for bowls, plates, knife handles and axes [55,112].

Despite the fragmentary composition of the samples collected from the corner Room G of the *Domus* of the *Dolia*, given the characteristics of this wood and the sampling place, a warehouse compartment, they were likely a part of the furniture or perhaps a tool present in the room.

Concerning the small number of fragments of maple wood sampled and identified at the *Domus* of the *Dolia* (Table 3), a plausible explanation is the characteristic of the easy perishability of this wood that would not have allowed for greater conservation.

With regards to the *P*. cf. *avium* wood charcoal fragments retrieved from the *Domus* of the *Dolia*, they likely belong to cherry wood. Wood charcoals of this species were recovered from Rooms G and H of the *Domus*.

Cherry is a hard, strong, heavy and elastic wood [55,107]. It is a precious wood especially appreciated for its colour and the possibility of being finished [40]. This wood can be easily attacked by insects, so it is not a suitable wood to be used outdoors, and it must be previously treated [40]. It is mainly employed for furniture, musical instruments, cabinetmaking and turning [40,107,113]. It is excellent for engravings and carvings [55]. The use of cherry wood for the production of objects has been known since prehistoric times, as well as its wide use for the manufacture and covering of furniture during Roman times [55,64,73]. Charcoals and woods of this species have been retrieved at the Bronze Age settlements of Via Neruda (Florence, Italy) [57,58] and in the Roman Villa of Poppaea (Naples, Italy) [83].

Samples of cherry wood from Room G of the *Domus* of the *Dolia* were retrieved in a fragmentary state, making their interpretation arduous. However, their sampling from the interior of one of the *Dolia* present in the room seems to suggest a possible interpretation as parts of the lid of the *Dolium*. With respect to the single charred wooden element of cherry wood retrieved from Room H of the *Domus*, we must take into consideration that Room H was a semi-open compartment, and that cherry wood is not suitable to be used outdoors. Considering that this wooden element was sampled in-between Rooms H and G, we may suppose that it was likely part of the dividing door of these rooms. However, we cannot rule out the hypothesis that it could also be a piece of a furniture or a tool present in the room.

*5.2. Court Trees*

Concerning the charred wood fragments of Rosaceae recovered from Room F of the *Domus* of the *Dolia*, it was not possible to distinguish between the various species of this family at the micro-anatomical level. This family comprises many fruit tree species of the Mediterranean area, for instance apple, pear and plum trees. Considering the archaeological context, wood charcoal remains of Rosaceae are likely to belong to a fruit tree/s. Room F was an open courtyard, and the retrieved wood charcoal remains of fruit tree/s allows us to hypothesize about the existence of a garden inside the *Domus*. Rosaceae wood charcoal fragments were the only *taxa* present in this room, accounting for 3.19% of the total wood charcoals identified (Table 3, Figure 6). This data seems to confirm the open-space nature of this room, in agreement with the archaeological data.

Regarding the small percentage of woods identified as possible court trees, this data can be explained by the burial conditions of these samples. Wooden remains of trees originally present in a garden—and therefore in an open-air space—have less chance of being preserved over time compared to wooden samples coming from a closed context.

*5.3. Exploitation of Different Local and Non-Local Tree Species and their Possible Supply Areas*

Everything seems to suggest that the wood choice was based on the characteristics of the woods, in order to employ them for different uses. However, cultural choices cannot be ruled out. The comparison of the identified tree species with the local vegetation (present and ancient) gave information regarding the employment of local and/or non-local species for the construction and furnishing of the *Domus* of the *Dolia*. The study shows, primarily, the employment of local tree species. The current vegetation of the Tuscan Maremma and its coastal hills, as well as the one where Vetulonia is located, is represented by sclerophyll and broad-leaved forests; there are holm oak woods with scrub and broad-leaved woods [114]. Regarding the species identified in this study, the three types of oak, *Q*. sect. *robur*, *Q*. sect. *cerris* and *Q*. sect. *suber* are considered local species, together with the tree species of *B. sempervirens*, *Acer* sp. and *P*. cf. *avium* [114,115]. This is confirmed by the results of pollen and charcoal analyses carried out in nearby areas of Vetulonia [10,14,15,17,116–124]. Many of these studies also revealed the strong impact made by the Etruscans on the Maremma

environment, as evidenced by the strong exploitation of some local species, including oak wood, as indicated by the decrease in its pollen in the historical record [10,14,17,122,123].

Concerning *A. alba* and *F. sylvatica*, indigenous populations of these two species currently naturally grow on the nearby Mount Amiata (approximately 60 km from Vetulonia) [125–127]. In this area, native fir populations are found at Pigellato (Piancastagnaio), Vivo d'Orcia and the Franciscan convent of the SS. Trinity of Santa Fiora [127]. The presence of silver fir wood on Mount Amiata in the past is proven by palynological data [128–131].

The possible supply of both species from Mount Amiata would therefore be supported by ecological data. There is also some historical evidence to support this hypothesis; Pope Pius II (1405–1464) employed the Amiata silver fir trees for the construction of his buildings in Pienza, and in his commentaries he wrote that the fir trees of Mount Amiata had also been used for the construction of ancient Roman buildings [132,133]. The supply of timber from this mountain appears to be confirmed by the ancient literary sources, which describe how the Romans usually obtained wood from this area of Etruria. The timber travelled on barges along the Ombrone and Albegna rivers, and, at the mouths of these rivers, port docks were placed; from there, the timber was transported to Rome [133,134]. The data from pollen and charcoal analyses [135,136] seems to attest that silver fir would have grown at lower altitudes than the current ones throughout the Italian peninsula in the early Holocene. It would grow at low and medium altitudes in forest communities associated with deciduous species, mainly *Quercus cerris*, at least until the Middle Ages. This data was associated with a more extensive presence of this species in Italy. The decline of this species seems to have mainly been caused by climate change and human impact [135]. According to this data, some historical literary sources attested to the presence of fir trees at Mount Amiata during the Etruscan and Roman period at lower altitudes than they are currently found [133].

*5.4. Consideration of the Technological Data*

Regarding the tree ring observations and the anatomical alterations recorded, it was possible to reconstruct, at least in part, the technological data. In this sense, only a few possible considerations are reported, and they should not be considered conclusive.

Concerning the fragments interpreted as wooden roof framing elements, it was not possible to identify a curvilinear trend in the tree growth rings, and therefore the fragments analysed come from trunks. This observation corroborates the archaeological interpretation of these samples. With regard to the samples identified as deciduous oak wood, tyloses were almost always observed in the lumen of the spring wood vessels. This means that the innermost part of the trunk (the heartwood) was used.

Given the fragmentary nature of many of the wooden roof framing elements identified, we don't know the thickness of the sapwood, and therefore it is not possible to know the thickness of the trunks from which the wood samples were obtained. The presence of bark was not detected for any of the samples under study, and it was not possible to detect the diameter of the complete stem, nor the season of tree felling.

Concerning the conservation state of the wood, the analysis enabled the identification of galleries formed by xylophagous insects in numerous charred wood fragments of the silver fir wood, and a few in the deciduous and semideciduous oak wood. These channels indicate a partial degradation of some structural elements of the *Domus* before the starting of the fire.

Regarding the samples identified as furniture and furnishing elements, the analysis made it possible to verify their good conditions at the time of the fire, as for most of them no signs of deterioration by wood-decomposing organisms were detected. Some beech wood fragments, in which channels formed by lignivorous insects were detected, are an exception, which indicates their partial degradation before the advent of the fire, but beech wood is not considered to be a very durable wood [137].

## 6. Conclusions

The study of the *Domus* of the *Dolia* woods is particularly important considering that studies on this type of material of the Etruscan era are very rare throughout the Italian peninsula. This is the first study concerning the construction and furniture woods retrieved from an Etruscan residence of this type. The information obtained from the analyses provided knowledge about the type of residence and occupants of the *Domus*. Moreover, the study can indirectly provide access to the economical, technological, and social indicators of the Etruscan community of Vetulonia, and more generally of the Etruscan society.

The study of charred woods from the *Domus* of the *Dolia* revealed the use of different types of woods employed for the construction of the roof and the furnishing of the *Domus*. The great variety of woods and their differentiated uses offered great insight into the wood choices of the *Domus* builders. The type of building, the quality, and the attention employed in the choice of woods suggests that the residence was inhabited by wealthy occupants.

The results also allowed us to establish the utilization of local and non-local woods. Furthermore, the observation of the anatomical characteristics provided evidence of the conservation state of the woods before the advent of the fire.

If we consider that most of our knowledge concerning Etruscan society comes from the study of materials from funerary contexts, this study on materials which come from a very well-preserved *Domus* (unique throughout the Italian peninsula) is like a drop in a sea of what is still largely unknown. The excavation of the *Domus* of the *Dolia* is still underway, and future research lines foresee the study of charred woods retrieved in the other rooms (the northern part) of the *Domus*.

**Author Contributions:** Conceptualization, G.C.; methodology, G.C.; formal analysis, G.C.; investigation, G.C.; data curation, G.C., S.R., C.Q.; writing—original draft preparation, G.C.; writing—review and editing, G.C., M.B.; supervision, C.B.D., L.S. All authors have read and agreed to the published version of the manuscript.

**Funding:** This research has been developed thanks to the HERITAS Doctoral Program through a PhD grant (FCT (PD/BD/128278/2017) founded by the FCT—Fundação para a Ciência e a Tecnologia. The authors also acknowledge the Hercules Laboratory project (UIDB/04449/2020 e UIDP/04449/2020) founded by the FCT.

**Institutional Review Board Statement:** Not applicable.

**Informed Consent Statement:** Not applicable.

**Data Availability Statement:** Not applicable.

**Acknowledgments:** The first author is grateful to Rebecca Mac Roberts for the English revisions, comments and useful advice.

**Conflicts of Interest:** The authors declare no conflict of interest.

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
