# Peer review of "The Wooden Roof Framing Elements, Furniture and Furnishing of the Etruscan Domus of the Dolia of Vetulonia (Southern Tuscany, Italy)"

_heritage, doi:10.3390/heritage4030110_

Round 1
Reviewer 1 Report
The paper consist an original and pioneering study on Anthracology in Etruscan culture. The work is well argued and structured.
Here explain some comments and suggestions that I feel need to be considered before to the publication.
Fig. 1: Increase the font for optimal display in A4. Also, I suggest using differently shaped symbols (and not color) for figure understanding when printing in black and white.
Line 196: Maybe, add "collected" before of the "wood charcoals".
Line 210: I suggest that the authors be more explicit about the size of the coal samples analyzed. In section "Materials", add a more detailed description of the size of the samples.
Line 214: This figure does not reflect the sampling strategy. Please, change figure caption.
Line 222: "the proportion was unlikely to have affected the overall interpretation". What efforts were made to sampling theses charcoals? Has the sediment floated? Why has it not affected the general interpretation? Please, argue this question further.
Line 228: Why was it not possible sometimes?
Line 254: Lack of precision about the method for the observation and analysis of the curvature of the growth rings. What method has been followed, what laboratory instruments?
Line 302: I suggest to change the X axis to the Y axis of table 3 so that the taxa names are not cut off.
Line 305: Regarding the presence of tyloses, why is this anatomical characteristic of oaks highlighted and not others? I suggest delete the phrase.
Line 561: I think the choice of wood for utilitarian purposes might be the most likely. But why is cultural choice ruled out? It could be implied.
In the results section, the growth ring curvature analysis results are missing.
Finally, I want congratulate the authors for the study and encourage them to continue in this line of research. Studies on wood-charcoal still have much to tell in protohistoric and historical cultures.
Author Response
Dear referee,
you can find the modified version of the manuscript in addition to the answer to your observations in the files attached below.
Best Regards,
Ginevra Coradeschi

Reviewer 2 Report
I suggest to accept it with minor revisions.
All the comments are in the attached PDF

Author Response
Dear Reviewer,
I am sending the answer to your comments. In the attached files below.
Thank you very much,
Best Regards,
Ginevra
